# ASPEN: Breaking Operator Barriers for Efficient Parallel Execution of Deep Neural Networks

**Jongseok Park**
Seoul National University
cakeng@snu.ac.kr

**Kyungmin Bin**
Seoul National University
kmbin@snu.ac.kr

**Gibum Park**
Seoul National University
gibumpark@snu.ac.kr

**Sangtae Ha**
University of Colorado Boulder
sangtae.ha@colorado.edu

**Kyunghan Lee**
Seoul National University
kyunghanlee@snu.ac.kr

## Abstract

Modern Deep Neural Network (DNN) frameworks use tensor operators as the main building blocks of DNNs. However, we observe that operator-based construction of DNNs incurs significant drawbacks in parallelism in the form of *synchronization barriers*. Synchronization barriers of operators confine the scope of parallel computation to each operator and obscure the rich parallel computation opportunities that exist across operators. To this end, we present ASPEN, a novel parallel computation solution for DNNs that allows *fine-grained dynamic execution of DNNs*, which (1) removes the operator barriers and expresses DNNs in dataflow graphs of fine-grained tiles to expose the parallel computation opportunities across operators, and (2) exploits these opportunities by dynamically locating and scheduling them in runtime. This novel approach of ASPEN enables **opportunistic parallelism**, a new class of parallelism for DNNs that is unavailable in the existing operator-based approaches. ASPEN also achieves high resource utilization and memory reuse by letting each resource asynchronously traverse depthwise in the DNN graph to its full computing potential. We provide challenges and solutions to our approach and show that our proof-of-concept implementation of ASPEN on CPU shows exceptional performance, outperforming state-of-the-art inference systems of TorchScript and TVM by up to $3.2\times$ and $4.3\times$, respectively.

## 1 Introduction

Deep Neural Networks (DNNs) are dataflow graphs of artificial neurons, each of which computes a mathematical function using the outputs of other artificial neurons as inputs. However, artificial neurons are rarely treated as individual units of computation, as their role as building blocks of neural networks has largely been replaced by *tensor operators*. A tensor operator, or simply an operator, refers to a large group of artificial neurons with the same functionality. These operators take inputs from the outputs of other operators, and the multi-dimensional arrays of data transferred between the operators are referred to as tensors. Modern DNN frameworks, such as TensorFlow [1], PyTorch[35], and MXNet[5], all utilize these operators as the main building blocks of DNNs, as grouping many identical artificial neurons into a single computation unit allows for easier construction, representation, and execution of DNNs that are becoming increasingly complex.

However, we find that the operator-based expression of DNNs incurs significant drawbacks in parallelism. Grouping artificial neurons into operators separates the computation of a DNN into two hierarchical layers of *inter-operator* and *intra-operator* computations[30, 51]. Inter-operator computations of a DNN are represented by a dataflow graph of operators, and frameworks such as Tensorflow or PyTorch delegate the execution of each operator to vendor-provided DNN acceleration libraries such as oneDNN [19], cuDNN [7], or ARMNN [3]. These libraries handle intra-operator

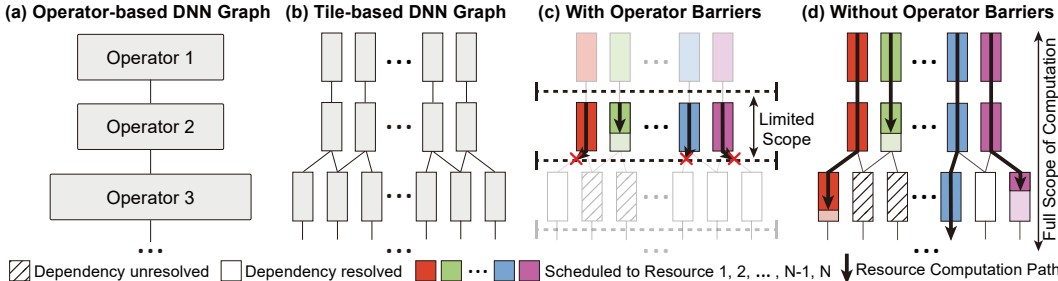

Figure 1: Depiction of a DNN as (a) operator-based dataflow graph and (b) tile-based DNN dataflow graph, and its execution using $N$ parallel computation resources, (c) with operator barriers, and (d) without operator barriers.

computations by partitioning the computations of an operator into fixed-shape computation units known as *tiles* [14, 28, 30, 54], that exploit key features of the hardware such as the number of registers, vector processing width, and cache sizes. For instance, oneDNN may partition single-precision matrix multiplication of $1024 \times 1024$ square matrices into $16384$ $8 \times 8$ matrix multiplication tiles, which align with the vector width of 256-bit AVX2 registers of x86 CPUs. The tiles are then scheduled to the parallel processing resources of the given hardware, and the execution of an operator is considered complete when all tiles have been computed by the parallel resources.

Unfortunately, this two-level execution of DNNs inevitably introduces a *synchronization barrier* between operators. As intra-level computations are treated as black boxes by DNN frameworks, synchronization barriers are necessary at the end of each operator execution to ensure that all computations within an operator are completed before scheduling the next operator that depends on its predecessor [45]. These barriers make the computation flow simpler and the development of execution frameworks easier, but they also completely obscure the rich parallel computation opportunities that exist *across* the barriers.

For example, Figure 1 (a) illustrates a traditional operator-based dataflow graph of a DNN at the inter-operator level. However, when we apply tile-wise partitioning at the intra-operator level and express the dataflow graph with tile-based granularity as shown in Figure 1 (b), we discover the presence of multiple parallel paths of computation across the operators. Unfortunately, the traditional two-level execution of DNNs depicted in Figure 1 (c) hinders the utilization of these computation opportunities, due to the synchronization barriers that confine the scope of computation within each operator. The synchronization barrier forces the resources to remain idle until all tiles within an operator are executed before allowing the execution of new tiles, resulting in an underutilization of available resources. A more efficient parallel execution could be achieved by removing the barriers and enabling each resource to asynchronously execute new parallel computations as soon as they become ready for computation, as depicted in the parallel computation paths of Figure 1 (d).

To utilize this untapped source of parallelism over the synchronization barriers, we propose *fine-grained dynamic execution of DNNs*, where we (1) remove the barriers and express DNNs in dataflow graphs of fine-grained tiles to expose the parallel computation opportunities across operators, and (2) exploit these opportunities by dynamically locating and scheduling them in runtime. This fine-grained dynamic execution of DNNs enables **opportunistic parallelism** [29, 26, 25] for DNNs, a new class of parallelism that is unavailable in the existing operator-based approaches. In opportunistic parallelism, each resource asynchronously traverses down a *distinct computation path* in the graph as depicted in Figure 1 (d). As there are no barriers to halt the execution of computation resources, each resource can execute its computation path to its maximal computational capabilities, leading to maximum system utilization and efficient load balancing. Also, as parallel resources are now computing down a path in the graph, data reuse is maximized as the computation output is reused as input for the next computation on each resource.

To fully leverage the potential of opportunistic parallelism in DNNs, we find three technical challenges that must be addressed. The first challenge lies in expressing the tile-wise dataflow graphs, from designing a partitioning approach that is general enough to be applicable to all DNNs, to determining the dimensions of the tiles that would allow the most efficient parallelism across the operators.

The second challenge involves developing a runtime system that can enable dynamic tracking of computation opportunities and asynchronous scheduling of many parallel resources over a complex DNN

dataflow graph. While the concept is straightforward, creating a parallel solution that achieves such asynchronous graph traversal and execution of many computation resources, without encountering race conditions or data hazards, while also maintaining high scalability and efficiency, is a formidable task. As existing operator-based frameworks can neither enable nor manage such asynchronous and dynamic execution of DNNs, a novel algorithm for DNN scheduling must be created to achieve efficient opportunistic parallelism.

The third challenge entails creating a solution that facilitates concurrent information exchange among massive numbers of asynchronous parallel resources. Even when each parallel resource executes an independent computation path, there inevitably comes a need for a resource to know the progression status of other nodes, for instance when a resource must select a new path to traverse. However, using synchronization for information exchange would halt the resources and compromise the effectiveness of opportunistic parallelism. Therefore, information exchange between resources must be performed asynchronously without impeding the progression of other resources.

To tackle these challenges, we present ASPEN, a novel DNN computation solution comprising three key components: (1) a **tile-based graph partitioning unit** that transforms operator-based DNN dataflow graphs into tile-based dataflow graphs unlocking rich parallel computation opportunities, (2) a **distributed scheduling algorithm** that enables each resource to asynchronously track and compute a distinct computation path without encountering any data hazards or race conditions, and (3) a **highly concurrent data structure** that facilitates asynchronous information exchange among parallel resources. These three components of ASPEN work in unison to address the aforementioned challenges and achieve efficient utilization of parallel computing opportunities across operators. Our proof-of-concept implementation of ASPEN on CPU demonstrates remarkable performance gains on various CNN and transformer-based model inference, achieving up to $3.2\times$ and $4.3\times$ speedup against state-of-the-art inference systems such as TorchScript [9] and TVM [6], respectively.

## 2 Background and Related Works

Limited parallelism of the current operator-based approach, particularly in the domain of *DNN inference*, has become a significant issue in recent years. In DNN training where an abundant number of inputs are provided, data parallelism or pipeline parallelism [8, 18, 31, 12, 47, 34, 51, 44] are extensively used to leverage the inherent concurrency between inputs and achieve highly-parallel DNN computation. However, in DNN inference, the number of inputs is often limited which restricts the parallelism available from concurrent input data [30]. To address this limitation, several solutions propose manipulating the operators of the DNN to expose more parallelism within the given DNN model structure.

One approach is **Operator Fusion**, which aims to merge computations from neighboring operators into a single, larger operator, creating a more substantial intra-operator computation space. This expanded computation space allows for greater parallelism opportunities, such as increased utilization of the vector processing hardware or batched memory accesses. Many high-performance DNN frameworks and acceleration libraries, such as TVM [6], TorchScript [9], XNNPACK [13] and oneDNN [19], have already integrated rule-based operator fusion and fused computation kernels to accelerate parallel DNN execution. Systems like TASO [22], Rammer [30], Apollo [49], and AStitch [52] introduce an advanced fusion technique which combines computation from independent operators into a single fused operator, unlike conventional fusions where only parent-child operators of a graph are fused. This technique can be understood as fusing inter-operator parallelism space into a larger, unified parallelism space, enabling much broader computation space for optimizations.

Another avenue to increase the parallelism opportunities is **Model Slicing** [50, 48, 17, 53, 21], which takes an opposite approach to operator fusion or stitching. Model slicing decomposes tensors and operators of CNNs into smaller ones, thus increasing inter-operator parallelism while reducing intra-operator parallelism. This approach is particularly utilized on edge clusters, where a large number of cluster nodes with limited computation capabilities synergy well with the increased number of operators and reduced per-operator computation.

Unlike existing works that focus on the manipulation of operators, ASPEN explores **tile-based dynamic execution of DNNs** for increased parallelism. Expressing DNNs in a tile-granularity effectively combines the separate inter- and intra-operator computation spaces into a single unified space of tile-based dataflow graph. This provides a holistic view of the DNN and enables a finer-grained

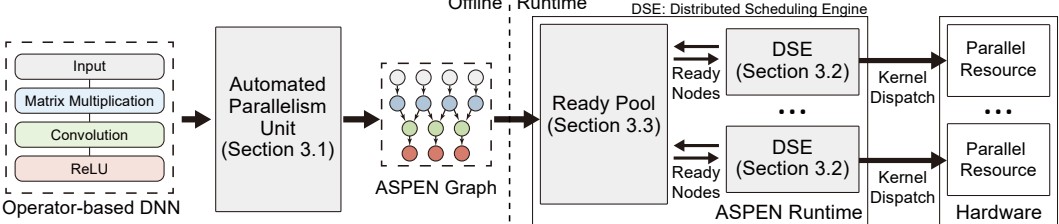

Figure 2: The overall workflow of ASPEN. APU compiles operator-based DNNs into ASPEN graphs to expose parallel computation opportunities across operators. ASPEN runtime, composed of Ready Pool and DSEs, utilizes opportunistic parallelism to achieve efficient parallel execution of DNNs.

analysis and management of both computation and data, allowing contributions that were previously impossible with the operator-based dataflow graphs. Recent works on tile-based understanding of DNNs focus on applying tile-based analysis and optimizations on topics such as DNN schedulers [30], graph compilers [54], or reducing memory overhead [40].

In contrast, ASPEN explores the benefits of tile-based DNNs *during runtime*. ASPEN combines (1) tile-based expression of DNNs, with (2) dynamic parallelism and execution approaches developed for irregular programs [29, 26, 25, 24, 15, 32, 33] to enable *fine-grained dynamic parallelism and execution of DNNs*. Finer granularity allows more parallelism opportunities to be expressed on the dataflow graph, and dynamic execution allows these opportunities to be located and scheduled right away during runtime, enabling a novel parallelism in DNNs which we call **opportunistic parallelism**.

To our knowledge, ASPEN is the first to explore the benefits of tile-based DNNs during runtime using dynamic scheduling and execution approaches. While we mainly focus on parallelism in this paper, we find that the benefits of fine-grained dynamic execution of DNNs are not limited to parallelism, and extend to dynamic load-balancing, reduced memory traffic, and novel functionalities that are impossible in operator granularity or static scheduling approaches. We cover these additional benefits in detail in Section 4 and 5.

## 3 ASPEN Design

To achieve efficient utilization of opportunistic parallelism in DNNs, we design ASPEN with three key components: the Automated Parallelism Unit (APU), the Distributed Scheduling Engine (DSE), and the Ready Pool. Each component is designed to overcome the three challenges described in Section 1, namely exposing computation opportunities across operators, creating an efficient runtime to leverage these opportunities, and enabling asynchronous information exchange between parallel resources. The following subsections provide a detailed explanation of how each component tackles its respective challenge.

Figure 2 depicts an overall workflow of ASPEN. The APU takes a traditional operator-based description of a DNN and automatically generates a computation tile-wise dataflow graph, which we call the ASPEN graph. The ASPEN runtime, which consists of DSEs and a Ready Pool, initializes the computation by loading the graph and input data. DSE is a scheduler that exists separately for each parallel resource and handles the asynchronous traversal and execution of parallel paths of the designated resource. As the execution progresses, information on the path progression of the DNN by each resource is updated in the Ready Pool as *ready nodes*. Ready nodes are nodes that have all their parent nodes computed but have not been computed themselves, and they represent the tail ends of execution paths. The DSEs can refer to the Ready Pool whenever they need information about other paths, enabling asynchronous information exchange between parallel resources.

### 3.1 Automated Parallelism Unit

The goal of APU is to transform an operator-based DNN into a tile-wise dataflow graph that exposes finer-grained parallel computation opportunities across operators. As mentioned in Section 1, existing computation kernels partition and execute operators using computation tiles [11, 43, 30, 54], with synchronization barriers to ensure the completion of all tile execution within an operator. APU removes these barriers and expresses dependencies between individual tiles to expose more parallelism. To create a generalized method applicable to any operator, we propose partitioning the

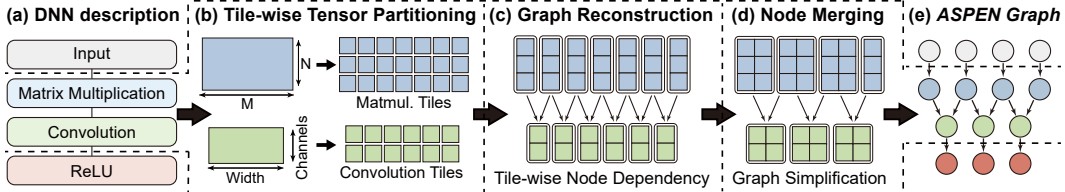

Figure 3: Illustration of the three-step APU operation on an example DNN.

output tensor of each operator into fine units to maximize parallelism and then merging them into graph nodes. Merging reduces scheduling overhead and allows for increased weight data reuse and better utilization of computation resources.

We observe that many DNN computations kernels are executed using matrix multiplication. Naturally, DNN computations form a chain of matrix multiplications where the output of one multiplication serves as the input for the next. We focus on the property that in matrix multiplication $A \times B = C$, only a single column-wise vector of matrix $B$ is required for the computation of the column-wise vector of matrix $C$ with the matching row index. If we fix the weight matrices as $A$, the chain of matrix multiplication in DNNs can be understood as a set of independent, parallel-running matrix-vector multiplication chains. Therefore, we conclude that partitioning operators into column-wise matrix tiles exposes the most parallel path within the DNN graph.

ASPEN leverages this insight by splitting output tensors into fine-grained matrix tiles aligned with the smallest hardware features such as SIMD register length or L1 cache size, and then merging them column-wise and subsequently row-wise. APU automates this process using a three-step approach illustrated in Figure 3 (b) to (d). First, (a) APU parses the given DNN and (b) partitions the output tensors into fine-grained matrix tiles. (c) It merges the tiles column-wise into graph nodes, creating a directed acyclic graph (DAG) based on the element-wise dependencies. (d) The performance of the resulting graph is evaluated, and nodes are further merged or split column-wise and then row-wise until a sufficient level of parallelism is exposed. (e) The resulting ASPEN graph now represents computation opportunities across operators as separate dataflow edges between nodes. The ASPEN graph is saved as a file and later loaded into the ASPEN runtime for execution.

## 3.2 Distributed Scheduling Engine

DSE aims to maximize the utilization of parallel resources by continuously scheduling new computation opportunities while dynamically traversing a barrier-free path in the DNN graph. Each parallel resource has its own DSE, operating in isolation to eliminate idling and loss of utilization due to synchronization. This decentralized approach distributes scheduling and graph overhead among the resources, enabling high scalability. We show our novel DNN scheduling algorithm ensures the correctness and completeness of DNN execution while operating in complete isolation.

In the ASPEN runtime, graph nodes transition between three states: *executed*, *ready*, and *not-ready*. An executed node is a node that has been processed by a computation backend after all of its parent nodes have been executed, or is an input (source) node of the DAG. A ready node has all its parent nodes executed but has not been processed itself. A not-ready node has one or more parents that are not executed. The state transitions always occur in the order of not-ready, ready, to executed.

DSE executes Algorithm 1. The idea of Algorithm 1 is that the DSE only needs to be aware of the ready nodes for its traversal, as ready nodes are always at the tail end of execution paths. Executing a ready node may turn one or more of its child nodes into ready nodes. If so, DSE selects one node for further traversal and stores the rest in the Ready Pool as new computation heads. If no new ready node is created, the DSE fetches a new path head from the Ready Pool. Algorithm 1 is also designed to be DNN-agnostic. That is, DSE will continuously fetch and execute new computations from the ready pool regardless of the layer or DNN graph it belongs to, to maximize resource utilization.

Figure 4 illustrates an example of DSE execution. When a DNN is loaded into the ASPEN runtime, the nodes from the first operator are always ready, as input nodes are always executed. These ready nodes are pushed into the Ready Pool to initiate execution. (a) DSEs of resources A and B fetch ready nodes as the heads of their execution paths. (b) Resource A reaches a dead end without any new ready nodes. (c) Resource A fetches a new path head from the pool, while resource B also hits a

**Algorithm 1** DSE's asynchronous graph traversal and scheduling algorithm

---

**Require:** Computation Resource $C$, Current ready node $N$, Ready Pool $R$
1: **struct** GRAPH NODE
2:     $K$ : COMPUTATION KERNEL OF THE NODE
3:     $A_c$ : ARRAY OF CHILD NODES
4:     $P_n$ : NUMBER OF PARENT NODES
5:     $P_e$ : NUMBER OF EXECUTED PARENT NODES
6: **end struct**
7: **while** True **do**
8:     **if** $N$ is NULL **then**
9:         $N \leftarrow pop(R)$                     ▷ Pop a new path head from pool. NULL returned if empty.
10:     **else**
11:         **Execute** $N.K$ on $C$       ▷ Execute current node. $N.X$ denotes the struct member $X$ of $N$.
12:         $a_c \leftarrow N.A_c$
13:         $N \leftarrow$ NULL
14:         **for** $n_c$ in $a_c$ **do**                  ▷ Iterate through all children of current node
15:             $p_e \leftarrow$ atomic_fetch_add$(n_c.P_e, 1)$         ▷ Atomic post-increment
16:             **if** $p_e + 1$ is $n_c.P_n$ **then**              ▷ If child is ready
17:                 **if** $N$ is NULL **then**
18:                     $N \leftarrow n_c$                ▷ Traverse to first readied child
19:                 **else**
20:                     $push(R, n_c)$         ▷ Push remaining readied children to pool
21:         **end for**
22: **end while**

Figure 4: Two parallel DSEs for Resource A and B executing Algorithm 1 on the example DNN from Figure 3. Resource A is assumed to be faster than B for demonstration purposes.

dead-end. (d) On its second computation path, resource A finds that both children of the executed node become ready. It proceeds with the first ready child while pushing the other child into the pool as a new computation head. (e) After all executions, the DNN computation is completed with five different computation paths taken by the resources. This novel execution enables asynchronous and continuous scheduling of new computation nodes to parallel resources, achieving highly efficient parallel execution of DNNs.

Algorithm 1 can be further optimized to certain DNNs or hardware. For instance, we find that dependency patterns in DNN graphs formed by pooling or strided layers create tiles of higher importance, and executions can be accelerated by prioritizing these tiles. Also, having a cache of child nodes on each DSE decreases the access to shared memory, which increases throughput. However, to focus on providing a general solution that first enables the novel approach of tile-based opportunistic parallelism, we leave optimizations for future work.

**Correctness:** We now show the correctness of the asynchronous parallel execution of ASPEN. We first show that race conditions are impossible. Let $v$ be a ready node. From Algorithm 1, a node is considered ready when $P_e = P_n$. Since the increment of $P_e$ is atomic, the DSE whose increment resulted in $P_e = P_n$ for $v$ is uniquely determined. The said DSE can choose to either set $v$ as its $N$ or push $v$ to the Ready Pool. If the former option is chosen, exclusiveness is guaranteed during the execution of $v$ as the DSE responsible for $v$ is unique. If the latter option is chosen, $v$ is stored in the Ready Pool until it is eventually popped by a DSE. As the Ready Pool is a concurrent data structure, only a single DSE can pop $v$, ensuring exclusiveness during the execution of $v$. Therefore, exclusiveness is guaranteed in all node executions of ASPEN.

Furthermore, all executed nodes of ASPEN yield correct execution output as long as a correct input to the DNN is provided. From Algorithm 1, a ready node $v$ becomes an executed node when it is

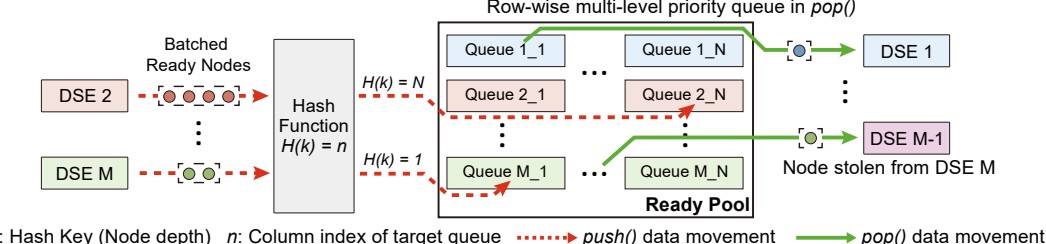

Figure 5: Illustration of concurrent access to Ready Pool queue matrix.

executed on a computation resource $R$ using kernel $K$. We assume $R$ and $K$ are correct, as they are externally supplied to the runtime. By the definition of a ready node, all parent nodes of $v$ are executed nodes. Since DSEs can only execute a ready node, no other DSE can modify the parent nodes of $v$, or the input of $v$, during the execution of $v$. This eliminates data hazards during execution and, when combined with exclusiveness in execution, guarantees correct execution results as long as the parent nodes of $v$ have correct execution results. Therefore, through recursion, ASPEN computes correct execution results for all nodes as long as a correct input to the DNN is provided.

**Completeness:** We show that all graph nodes become executed in ASPEN. Suppose there exists a node $v$ in the DAG that is never executed. $v$ cannot be a ready node since a ready node will be either executed as soon as it is created, or stored in the Ready Pool until it is eventually popped and executed. Therefore, $v$ must be a not-ready node, and by definition of a not-ready node, at least one parent of $v$ is also a node that is never executed. By recursion, there must exist a path from some source node $s$ to $v$ where all nodes on the path are not-ready nodes. However, this leads to a contradiction as source node $s$ is an executed node by definition. Therefore, $v$ cannot exist in ASPEN.

### 3.3 Ready Pool

Ready Pool is our fast, flexible, and scalable solution for managing dependencies among a large number of computation nodes and parallel resources. It acts as a barrier, separating nodes that can be computed from those that cannot, similar to existing synchronization barriers in operator-based approaches. However, Ready Pool offers a significant advantage over traditional synchronization barriers in that the dependency information exchange happens on-demand, and only between the producer and consumer of the information, without involving other resources.

Synchronization barriers inserted in compile time are unaware of which computation tile is scheduled to which resources on runtime. As a result, they must synchronize all resources to ensure the correctness of dependent computations, which limits parallelization scope and hampers the utilization of faster resources. In contrast, Ready Pool allows resources to dynamically update the satisfaction of dependencies to the pool and enables asynchronous retrieval by other resources when they require new computations, minimizing the overhead of data exchange.

However, if not properly designed, Ready Pool can be a bottleneck in the system. To ensure constant access time, flexible scheduling policies, and scalability over many parallel resources regardless of DNNs used, we design Ready Pool using a matrix of concurrently accessible FIFO queues, as shown in Figure 5. Each DSE is assigned a row of queues, and accesses from a DSE are prioritized within its assigned row to minimize conflicts between DSEs. A simple hash function determines the column index of the accessed queue during the $push()$ operation, while a multi-level priority queue is used during the $pop()$ operation to facilitate fast accesses and support scheduling policies.

To be specific, during $push()$, ready nodes from each DSE are batched to reduce overhead. The target queue's column index is determined using a user-defined hash function $H(k)$. In our proof-of-concept code, key $k$ is the depth of the pushed nodes, and $H(k)$ returns a smaller column index if the depth is shallow, and a larger index if the depth is deep. When combined with row-wise priority queue access of $pop()$, this implementation enables the scheduling policy of prioritizing nodes of shallower depth, which accelerates execution by allowing DSEs to traverse longer computation paths. During $pop()$, a DSE first pops nodes from its assigned row to maximize data reuse. If the assigned row is empty, the DSE searches rows of other DSEs similarly to work-stealing queues [4], to automatically load balance and improve resource utilization. When a DNN enters the ASPEN runtime, ready nodes are

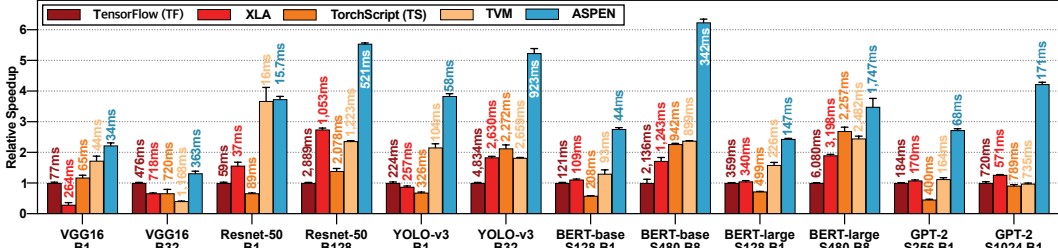

Figure 6: Relative latency speedup and latency of ASPEN and other frameworks compared to TensorFlow [1] on Threadripper 3990X. Latency measurements are annotated on each bar. B refers to the batch size, and S on the NLP models refers to the number of input tokens.

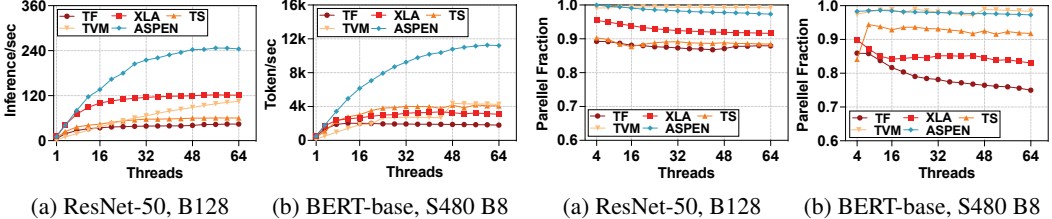

(a) ResNet-50, B128    (b) BERT-base, S480 B8    (a) ResNet-50, B128    (b) BERT-base, S480 B8

Figure 7: Throughput scaling of ASPEN and other frameworks over Resnet-50 and BERT-base using the full core range of Threadripper 3990X.

Figure 8: Experimental parallel fraction $p_e$ over Resnet-50 and BERT-base using the full core range of Threadripper 3990X. Higher is better.

uniformly distributed among the rows. This design of Ready Pool ensures fast and concurrent access while enabling scheduling policies and automatic load balancing throughout the DNN execution.

## 4   Evaluations

**Implementation Details:**   As no existing DNN framework supports the use of opportunistic parallelism, we implement our proof-of-concept code of ASPEN targeting CPUs with approximately 12k lines of C code from scratch. We create our own tile-wise GEMM kernels using AVX2 extensions and conv2D kernels using the GEMM kernels and im2col. The remaining operators are computed using simple C for-loops. For memory, output tensors for each operator are created in NHWC order as in the existing approaches, and each tile holds a pointer to its respective location in each tensor. Our implementation is publicly available at https://github.com/cakeng/ASPEN/tree/ASPEN_NeurIPS/.

**Experimental Setup:**   Our evaluations are conducted on an AMD Threadripper 3990X 64-core processor and an Intel i9-12900K 16-core processor using Ubuntu 22.04. We compare ASPEN against the popular DNN framework of TensorFlow (v2.7) [1] as well as highly-optimized inference solutions of TorchScript (v2.0.0) [9], TensorFlow XLA (v2.6.2) [39], and TVM (v0.11.0) [6], all using C/C++ API. We evaluate inference latency of various CNN and Transformer-based [46] NLP models, namely VGG-16 [42], ResNet-50 [16], Yolo-v3 (416) [38], BERT-base, BERT-large [10], and GPT-2 (124M) [37]. For the GPT-2 model, we evaluate the first model iteration, where no past attention values are provided. We measure the end-to-end latency of DNN model execution, which excludes pre- or post-processing such as image cropping or text tokenization. We average the measurements over 100 runs to obtain representative values.

### 4.1   Execution Latency

Figure 6 presents the inference latency speedup of ASPEN and other frameworks compared to TensorFlow on Threadripper 3990X for various DNN models. Overall, ASPEN demonstrates strong performance, achieving speedups up to $6.2\times$ against TensorFlow (BERT-base S480 B8) and $4.3\times$ against TVM (GPT-2 S1024 B1). We find ASPEN performs better when more layers with multiple execution paths, such as residual connections or multi-head attention layers, are contained within the model. These network designs provide multiple computation paths across operators, which synergize effectively with the opportunistic parallelism of ASPEN.

ASPEN also exhibits amplified speedup with larger batch sizes, as larger batches further facilitate the isolation of computation between DSEs. In existing operator-based solutions, dependent compu-

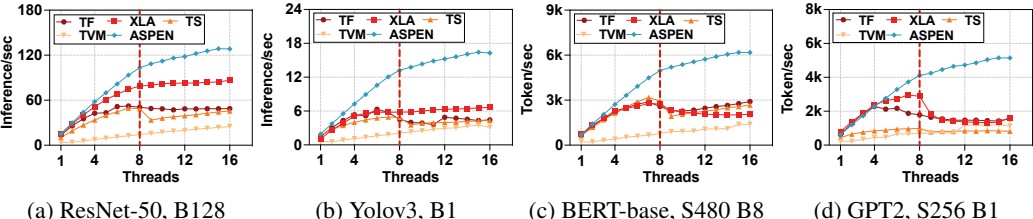

(a) ResNet-50, B128    (b) Yolov3, B1    (c) BERT-base, S480 B8    (d) GPT2, S256 B1

Figure 9: Heterogeneous resource utilization of ASPEN and other frameworks over various DNNs on Intel i9-12900K. B refers to the batch size, and S on the NLP models refers to the number of input tokens. Both the performance cores and efficiency cores are utilized after 8 threads.

tations across operators are not guaranteed to be scheduled to the same resource. This necessitates scatter/gather-like data exchanges between resources causing memory overheads, and this overhead only increases with larger batch sizes and more resources used. In contrast, depth-first computation of ASPEN allows dependent tiles to be scheduled to the same resource as much as possible, achieving effects similar to operator fusion. With large enough batch sizes, each DSE is allocated paths in different batch indexes similar to data parallelism, greatly reducing the memory overhead. Also, even when there are data exchanges between resources, only a few resources are involved simultaneously, relieving the pressure on the memory system.

## 4.2  Parallel Scaling

Figure 7 presents the strong scaling throughput results of ASPEN against other frameworks on Threadripper 3990X, detailing the results of Figure 6. We observe that ASPEN exhibits superior per-resource scaling performance over other frameworks, thanks to its asynchronous design and distributed scheduling. To quantitatively evaluate the scaling performance, we introduce the experimental parallel fraction, denoted as $p_e$, in Figure 8. $p_e$ represents the proportion of computation executed in parallel, which directly influences the scaling and upper limit of parallel speedup according to Amdahl's law [2]. We use Karp-Flatt metric [23] $p_e = 1 - \frac{1/\psi - 1/N}{1 - 1/N}$ to calculate $p_e$, N being the number of parallel resources, and $\psi$ being the measured speedup while using N parallel resources. ASPEN achieves remarkably high $p_e$ values ranging from 0.97 to 0.99 across all cases, indicating that more than 97% of ASPEN computations are performed in parallel, which allows ASPEN to exhibit exceptional scaling performance following Amdahl's law.

## 4.3  Resource Utilization

Figure 9 presents the resource utilization of ASPEN and other frameworks on Intel's i9-12900K heterogeneous processor for various DNN models. i9-12900K has 8 performance cores (P-cores) and 8 efficiency cores (E-cores) with different computing capabilities. We observe that ASPEN shows a linear summation of all utilized core performance, clearly highlighting the higher performance inclination of P-cores on 1 to 8 threads and the lower performance inclination of E-cores on 9 to 16 threads. In contrast, existing solutions exhibit sub-optimal resource utilization when using both the P-cores and E-cores, suffering from stale performance increases (TensorFlow, XLA), sharp performance drops (TorchScript, XLA), or performance bottlenecks to the slower E-cores (TorchScript, TVM). ASPEN, on the other hand, effectively utilizes all available resources to their full potential, thanks to its dynamic scheduling and automatic load-balancing capabilities provided by the ASPEN runtime.

## 4.4  Ablation Studies

Figure 10 (a) provides the execution throughput of ASPEN in FLOP/s against the number of ASPEN tiles per layer. The throughputs of other frameworks are presented in dotted horizontal lines. The throughput of ASPEN increases on 1 to 128 tiles, as increasing the number of tiles provides more parallelism. The throughput drops after 128, as there are not enough resources in the machine to utilize the increased parallelism, while the smaller tiles increase overhead and reduce computation efficiency. As such, ASPEN shows a concave performance characteristic against the number of tiles.

Figure 10 (b) to (d) provides the execution throughput scaling of ASPEN and other solutions against various DNN parameters, normalized to the throughput on the smallest parameter size. ASPEN shows

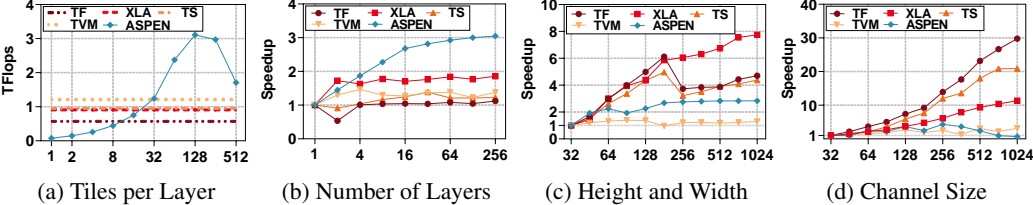

| (a) Tiles per Layer | (b) Number of Layers | (c) Height and Width | (d) Channel Size |

Figure 10: Performance and scaling against differing parameter sizes, on Threadripper 3990X using 64 cores. All tests are executed with a batch size of 1, using a synthetic DNN with 32 identical convolution layers. All layers have 64×64 inputs, 3×3 filters, input and output channels of 128, padding and stride of 1, and a fixed 128 tiles per layer for ASPEN, except for the specified parameters.

a noticeable speedup with an increasing number of layers, owing to the utilization of computation opportunities and depthwise computation across the layer boundaries. Against height, width, and channel sizes, existing frameworks except TVM exhibit large performance speedup as larger tensor sizes allow more parallelism in each layer for these solutions. The performance of TVM is largely unaffected as it creates optimized kernels for each of the given parameter sizes, achieving constant performance across tensor sizes. The performance of ASPEN is also less affected by tensor sizes as ASPEN is able to source its parallelism from other sources.

## 5   Limitations and Discussions

**ASPEN on GPUs:** While our current proof-of-concept implementation of ASPEN targets CPUs, we expect ASPEN to be easily applicable to GPUs as well if given proper tile-level kernel support. For example, Nvidia GPUs can leverage the CUDA Streams and CUDA Events API to dispatch tile-level kernel calls asynchronously, while DSEs on the CPU perform graph traversal to identify computation opportunities. Using ASPEN on GPUs would allow for a seamless interleaving of data movement, kernel launches, and tile executions. This would greatly reduce the host-device scheduling and communication overhead, which are often reported as a limiting factor of GPU utilization [30, 27]. The asynchronous nature also means that the computing resources would be at differing stages of kernel execution, which distributes memory access requests across the temporal domain and mitigates the limitations in memory bandwidth that DNN executions on GPUs often face [51, 40].

**ASPEN on DNN training:** ASPEN can potentially be used for DNN training as it offers a general solution for leveraging opportunistic parallelism on DNNs. However, as DNN training can exploit the abundant input data as an alternative source of parallelism, it is unlikely that ASPEN's dynamic approach would outperform static optimization solutions [41, 36] due to runtime overhead. Nonetheless, ASPEN remains appealing in environments with varying capabilities, such as edge computing.

**ASPEN for diverse applications:** Fine-grained dynamic DNN execution of ASPEN also brings several novel functionalities to DNN execution. Since the execution of DSEs is DNN-agnostic, different DNNs can be co-executed easily for increased system utilization [20] by simply placing them in the same Ready Pool. Parallel resources can be dynamically added or removed from the ASPEN system without disrupting the execution of other resources, allowing for enhanced flexibility. For DNN applications that operate on continuous input streams like videos, ASPEN's dynamic dependency tracking enables the execution of only the relevant tiles affected by the changes in the input, significantly reducing computation requirements. In computation offloading, ASPEN can interleave computation and transmission at the tile level to hide most of the networking overhead. In inference servers, inference requests can be immediately pushed into the Ready Pool and executed concurrently with other users' requests, reducing turnaround time and improving system utilization.

## 6   Conclusion

We present ASPEN, a novel parallel computation approach for DNNs that aims to (1) eliminate the synchronization barriers of tensor operators and (2) leverage opportunistic parallelism on the tile-wise dependency graph of DNNs. This allows ASPEN to dynamically locate and execute any parallel computation opportunities, resulting in high scalability and efficient utilization of parallel resources. Through evaluation, we validate that ASPEN outperforms the existing solutions and delivers exceptional parallel computing performance for DNNs.

## Acknowledgments and Disclosure of Funding

This research was supported in part by the IITP grant (2022-0-00420) and National Research Foundation of Korea (NRF) grant No. 2021R1A2C2006584 and No. 2022R1A5A1027646, funded by the Ministry of Science and ICT (MSIT) in Korea. Kyunghan Lee is the corresponding author.

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
