# OpenReview forum: "ASPEN: Breaking Operator Barriers for Efficient Parallelization of Deep Neural Networks"
_NeurIPS.cc/2023/Conference — NeurIPS 2023 poster_

### Official Review · Reviewer_Cexe · 2023-06-26

**Soundness:** 3 good
**Presentation:** 3 good
**Contribution:** 3 good
**Rating:** 6
**Confidence:** 4

**Summary:**

The authors proposed ASPEN, an opportunistic parallelism method that breaks the synchronization barriers of each operator presented in a DNN graph so that parallel compute resources can tranverse and execute multiple data-paths independently with much less synchronization overhead in a **shared memory** system. This is achieved by:
* Split the operators/tensors to multiple tiles thus multiple data-paths along the dataflow graph
* Implements **decentralized** executors to tranverse these data-paths indepently with only in-demand synchronization with a **centralized** information pool that captures the dependencies of the split graph

**Strengths:**

* The authors explored opportunisitic parallelism, a less studied/adopted parallelsm in modern machine learning frameworks/infrastructures, which is particularly helpful when three scenarios are present, namely:
  1.  The hardware system has significant threading (barrier overheads) like CPUs
  2. The ML models enbrace multiple data-paths (like residual connecitons or multihead attention)
  3. The input data has limited fully independent dimension like batch dim during inference

* The implementation of ASPEN is intuitive/efficient with correctness/completeness proves

* The evaluation of ASPEN on Resnet and Bert demonstrated its effificacy (significant speedup) compared to popular frameworks implemented with operator barriers.

* ASPEN could be helpful on inference environment with heterogeneous/edge devices

**Weaknesses:**

* While the authors explored opportunisitic parallelism and demonstrated its efficacy on a single CPU with multi-cores, its application is limited as the three scenarios mentioned in Strengths hardly coexist in modern ML workloads, e.g,.
  1. GPUs don't have such high threading overheads, thus has much higher intra-operator parallel efficiency. GPU systems are usually bottlenecked by device memory bandwidth or interconneciton bandwidth, which can't be solved by opportunisitic parallelism.
  2. When the app has abundant input data (e.g., large batchsize) like in training or batched inference, ASPEN is not likely to provide performance margin over other frameworks (line 328)

* ASPEN is only valid for a shared memory system, e.g, multi-threading on a single CPU, when scaled to multiple CPUs or multiple hosts (e.g., multi-processing), the IPC or interconnections could introduce significant memory overhead

* Some of the implementations are not very clear, e.g,.
  1. in line 176, the authors metioned the graph is split/merged column-wise and row-wise, but on line 178 it said "It merges the tiles column-wise" and nothing is mentioned for row-wise, so is in Figure 3.
  2. The explanation of Ready Pool in section 3.3 is not very intuitive. On a high level I understand it tries to prioritize ready nodes with shallower depth and data reuse, but I am not sure why it has to be a matrix with hashing to column indices, e.g., why is a row-wise priority queue not enough
  3. The authors didn't explain how memory allocation/deallocation is handled, e.g., is splitting operatos results in memory fragmentation? When is the time to garbage collect and who handles it, etc.

* Evaluations are not comprehensive or a bit contradictory to the claims:
  1. ASPEN should provide more values when the input size is small (line 109/328) compared to other frameworks, however Figure 6 shows that ASPEN demonstrates higher speedup when input/batchsize is larger, and authors attributed to larger inputs faciliate an increased number of isolated computation paths across operators, however, other frameworks should also benefit from it as it amortizes barrier overhead (line 328)
  2. Related to 1, therefore an ablation study on how batch/sentence/image size affects ASPEN should be provided to understand where ASPEN really outperforms other platforms. Addtionally since the authors mentioned models with more execution paths are likely to benefit (line 290), it is worth breaking down ASPEN formance on MHA and MLP(FFN) of a transformer layer to see which provides more performance gain. Other than MHA, artificial graph with various execution pass may also serve as a good ablation study.

**Questions:**

In addition to the questions mentioned in Weakness, here are a few more:
1. As author mentioned, column-wise (aka on batch-size) graph partitions typically expose most parallel paths (line 173) that doesn't require synchronization with the pool. But do you also partition row-wise/image-wise? Since these partitions may incur reduce-scatter/all-gather operations (even batch-partitioned Resnet have similar issue due batch-norm), are they handled by ASPEN? Even if it is a shared memory system, would it introduce high overhead? And if so, how do ASPEN decide which dimension to tile and to what degree?

2. in line 281, the authors mentioned GPT2 is only evaluated for prompt/question encoding which usually has large sequence length. However the majority of a generative inference task is a per token infernece where Batchsize = SequenceSize = 1, how is ASPEN's performance on this? have you considered head tiling to speed it up?

**Limitations:**

The authors have discussed the limitations which are plausible, however there are still a few more to address:
1. Efficacy on GPUs. The authors mentioned similar ideas can be implemented with CUDA Streams and Events, however as mentioned earlier, GPUs don't have such high threading overhead, so its efficacy is quesitonable compared to un-tiled but well vectorized GPU kernels.
2. Though the implementaiton of DSE and Ready Pool looks robust, due to it is still a runtime/software managed data dependency controller, its efficiency can't be compared to a native dataaflow architectures with hardware managed data dependency, e.g, IPU/RDU/WSEs from GraphCore/SambaNova/Cerebras.
3. As authors suggested, ASPEN can improve system utilization when shared with users, however this comes at the cost of higher per user latency even though overall system utilization can be higher.
4. As is mentioned in weakness, it is limited to a shared memory system so can't be leveraged in a distributed system.
5. The authors said ASPEN is orthogonal to existing fusion and slicing techniques (line 138), however, fusion and slicing would either amortize barrier overhead or fight for slicing/splitting degrees with ASPEN, which diminishes the performance gain of ASPEN.

Nevertheless, ASPEN still shines on the exploratory works it has conducted for opportunisitic parallellism in ML, despite all the limitations above.

---

> ### Author Rebuttal · Authors · 2023-08-09
>
> Thank you for your valuable comments and feedback! In this response, we will address and clarify your individual concerns one by one.
>
> ___
>
> > **Issue 1** Applications of ASPEN are limited.
>
> * GPUs don't have high threading overheads and are bottlenecked by memory bandwidth
>
> Overcoming the threading overheads is one of the main contributions of opportunistic parallelism, but its dynamic and asynchronous nature provides additional benefits. We explain the detailed benefits of ASPEN on GPU in the general response.
>
> * ASPEN is not likely to be useful with abundant inputs.
>
> As shown in the evaluations, this is clearly not true. We explain this in detail in Issue 4.
>
> > **Issue 2**  ASPEN does not scale to multiple hosts.
>
> We are currently extending ASPEN to a multi-host system as a separate work. We first separate the ASPEN graph into subgraphs that belong to different hosts. Once a DSE computes a node, it checks if its child belongs to a different host, and sends its data to the child’s host if so. The receiving host marks the node as complete and updates its child nodes. A child node enters the ready pool only when it belongs to the current host. ASPEN also brings benefits to such multi-host scenarios as the host computations and data transfers are effortlessly interleaved with one another, allowing for high utilization of both the hosts and the network.
>
> This is clearly beyond the scope of proposing the concept and implementation of opportunistic parallelism in this paper, but to provide an insight into the potential of ASPEN, we will include a discussion on the multi-host extension.
>
> > **Issue 3** Some implementations are unclear.
>
> * Row-wise split/merge is missing on line 178 and Figure 3.
>
> Row-wise split/merge happens in stage (d) of the given example. We will make this explicit.
>
> * Why is a row-wise priority queue not enough?
>
> The hashing and priority queue is combined to facilitate fast access times. By using hashing and priority queues, we can place shallower/deeper nodes in higher/lower priority queues in constant time and retrieve them in constant time. If we used sorted structures such as heaps, the access times would have scaled to the number of nodes, which would possibly be a bottleneck in the system.
>
> * How is memory handled?
>
> Nodes of ASPEN graphs act only as instructions to compute the given tile in a tensor. As such, it only holds pointers to its respective tile location. These tile locations are still a part of the original input/output tensors, as in the conventional operator-based approach. As such, no fragmentation exists, and memory allocation works in the same way in existing solutions. While we currently do not have runtime memory management, its implementation is trivial. Tensors are allocated when one of its tiles becomes ready and is deallocated when all dependent tiles of the tensor are computed. The allocation is handled by the DSE which updated the tensor’s first readied tile, and deallocation by the DSE which computed the last dependent tile.
>
> We will include the above explanations in our final manuscript. We also plan to fully release the ASPEN source code, so any details of its inner workings will be transparent.
>
> > **Issue 4** ASPEN is faster with a larger input size, which is not comprehensive to the claims. Ablation studies are needed.
>
> Increased batch sizes benefit ASPEN execution as it enables much greater data isolation between resources. In a traditional two-layered operator-based computation, it cannot be guaranteed that dependent computation tiles will be scheduled to the same parallel resource, as each operator computations are separate function calls. This requires scatter-gather memory operations between the parallel resources, which causes increased memory overhead and loss of computation utilization, whose impact increases as the number of parallel resource scales.
>
> Depth-first computation and the Ready Pool allow dependent tiles to be scheduled to the same resource as much as possible, achieving effects like that of operator fusion. This increases data reuse and reduces memory traffic between resources. With large enough input sizes, it becomes possible that each resource never has to share its outputs with other resources, as each DSE is automatically allocated paths in different batch indexes, greatly reducing the memory overhead. Also, even when there are data shared between resources, only a few resources are involved simultaneously, which relieves the pressure on the memory system.
>
> In our final manuscript, we will add a clear and explicit explanation of what isolated computation paths from large input sizes mean.
>
> We also agree that ablation studies such as varying input sizes, and sub- or artificial-graph level experiments can help understand the speedup of ASPEN. However, contributions of ASPEN such as increased parallelism over operator boundaries, or asynchronous interleaving of scheduling and computation are most prominent in an end-to-end execution and are not easily visible in microbenchmarks. Nevertheless, we plan to add the mentioned evaluations.
>
> > **Issue 5** Partitioning row/image-wise would incur high overhead.
>
> As explained in Issue 3, tiles are referenced as pointers, and the tensors are not actually partitioned in memory. Therefore, scatter/gather or other memory operations that are not present in operator-based computations are unnecessary.
>
> > **Issue 6** How is ASPEN’s performance on per-token inference?
>
> As per-token transformer inference is similar to a single chain of matrix-vector multiplications, there is limited room for opportunistic parallelism, and thus the performance is quite limited. However, additional optimizations such as head tiling or iteration-level scheduling proposed by Orca (Yu et al., OSDI 2022), can be integrated for increased performance. We believe that these techniques would bring novel and intriguing prospects when combined with fine-grained dynamic scheduling and execution of ASPEN.

---

> > ### Comment · Reviewer_Cexe · 2023-08-13
> >
> > Thanks for the detailed explanation. I have a couple further questions:
> >
> > 1. Issue 1. I am convinced that same ideas can be applied to GPUs through tiled kernels/CUDA event apis, though my concern is when GPU is already compute (large batch-size/input, pretraining) or memory bound (generative inference) rather than synchronization/kernel-launch bound, how much value can ASPEN adds? This may not be a fair question since ASPEN is an exploratory work on a new parallelization regime, but its benefit can be underestimated when the industry is dominated by LLM workloads.
> >
> > 2. Issue 5. Now I got tiles are just pointer offsets, which makes sense in terms of performance and memory management, though I am still unclear if you would partition along a reduction/accumulation dim (which would introduce an all-reduce in distributed system), and if so how do you evaluate the overhead of the induced accumulative operations and how do you decide which dimension to partition.
> >
> > Addtitionally I believe Limitation bullet points 2 and 5 are not addressed.
> >
> > Regardless, thanks again for your contribution and I am looking forward to your final manuscript.

---

### Official Review · Reviewer_yggz · 2023-06-26

**Soundness:** 2 fair
**Presentation:** 3 good
**Contribution:** 2 fair
**Rating:** 6
**Confidence:** 3

**Summary:**

When we run a deep neraul network, a sequence of operators are executed. The existing deep learning framework/compiler would wait for the completion of the prior operator (A) before launching the subsequent one (B). The authors of this paper observe that some computation in operator B only depends on part of the computation of operator A. Thus, the authors propose to decompose the operators into more fine-granularity and utilize the fine-grained dependency to expose more parallelism (they called it opportunitic parallelism). They implemented this idea and experiments show that it achieves up to 3.2x to 4.3x speedup compared with prior work TorchScript and TVM, respectively.

**Strengths:**

1. The authors observed the unnecessary barrior between operators and observe a new kind of parallelism.
2. The authors implemented a prototype system (ASPEN) and experiments show that the new parallelism is benefitial to CPU inference on a multi-core CPU.


**Weaknesses:**

1. More case study is needed to justify where the speedup comes from.

From Figure 6, we can see that ASPEN can achieve 4x (GPT-2 S1024 B1) and 2x (ResNet50 B128) speedup. I am quite interested in where the speedup come from. Usually, a new kind of paralleism (in this case, opportunistic parallelism) can have a good speedup when the existing parallelism is not enough to saturate the device. For example, parallelizing two operators would have a good speedup if each operator is not large enough to fully-utlize the device. However, the operators in the two models with large sequence length or batch size should be already large enough to highly utlize the device (rely on intra-operator parallelism), then how can the opportunistic parallelism gain more speedup? Does other factors contribute to the speedup? For example, C++ implemented runtime, statically allocated memory (instead of dynamic allocated as in PyTorch), more efficient kernels for other operators like batch norm, group norm, and softmax, better micro kernels for gemm. Thus, I would suggest to add more case study to decompose the speedup. For example, do some operator-level and sub-graph-level experiments, use profilers to analyze the CPU utilization, use ASPEN's microkernels but do not use opportunitic parallelism (e.g., keep the operator barrier) to get the performance change.

BTW, I have tried the ResNet50 example and observed good speedup on my workstation.

2. More dtails for ASPEN implementation and baselines are needed

For the TVM baseline, what scheduler have you used: AutoTVM or AutoScheduler (Ansor)? What tuning configuration did you choose? For ASPEN, did you write all kernels by yourself, or based on any existing BLAS library?



**Questions:**

1. Could you please provide more case study to get the contributing factors of speedup?
2. Could you elaborate more details of the ASPEN implementation?

[Minor]

It's better to also cite and compare with Graphi[1], Nimble [2] and IOS [3] in Section 2 when discuss inter-operator parallelization.

- [1] Graphi: Scheduling Computation Graphs of Deep Learning Models on Manycore CPUs
- [2] Nimble: Lightweight and Parallel GPU Task Scheduling for Deep Learning
- [3] IOS: Inter-Operator Scheduler for CNN Acceleration


**Limitations:**

1. Currently, only validated the effectiveness on CPU.

---

> ### Author Rebuttal · Authors · 2023-08-09
>
> Thank you for your valuable comments and feedback! In this response, we will address your individual concerns one by one in addition to the general response. We hope this clarifies your concerns about our work.
>
> ___
>
> > **Issue 1** More case study is needed to justify where the speedup comes from.
>
> We agree that ablations studies on the operator or sub-graph level would be helpful to further understand the speedup of ASPEN. However, it must be noted that contributions of ASPEN such as increased parallelism over operator boundaries, asynchronous interleaving of scheduling and computation, or data reuse from depth-first execution are most prominent in a full end-to-end execution and are not easily visible in microbenchmarks with a limited number of operators or computations. Nevertheless, we plan to add kernel-level performance evaluations and memory and CPU utilization profiling results to our final manuscript, to more clearly show where the benefits of ASPEN come from.
>
> For the increased speedup in large input sizes, we attribute the cause of the speedup to the isolation of each computation resource in ASPEN. In a traditional two-layered operator-based computation, it cannot be guaranteed that dependent computation tiles will be scheduled to the same parallel resource during subsequent operators, as operator calls are implemented as separate function calls. As such, scatter-gather memory movements between resources are required to share their computation data, which causes increased memory overhead and loss of computation utilization, whose impact increases as the number of parallel resource scales.
>
> Depth-first computation algorithm of DSEs and the two-dimensional Ready Pool allows dependent computation tiles to be scheduled to the same computation resource as much as possible, achieving effects like that of operator fusion, but in an implicit way. This reduces memory traffic between resources and allows data to be reused as much as possible. With large enough input sizes, it becomes possible that each resource never has to share its computation outputs with other resources, as each DSE is automatically allocated computation paths with different batch indexes, greatly reducing the memory overhead of the computation. Also, it should be noted that even when there are data shared between resources, only a few resources are involved simultaneously, which relieves the pressure on the memory system.
>
> In our final manuscript, we will add a clear and explicit explanation of what isolated computation paths from large input sizes mean, and how it improves the execution of ASPEN in terms of scalability, data movement, and resource utilization.
>
> > **Issue 2** More details for ASPEN implementation and baselines are needed.
>
> We will include more implementation details such as memory layout and access patterns of ASPEN, as well as details on computation kernel implementations and evaluation environments in our final manuscript. We also plan to fully release the ASPEN source code, so any details of ASPEN and its inner workings will be very transparent.
>
> For the TVM baseline, we used AutoTVM following the instructions provided on the official TVM documents. We compiled the TVM programs in Python and exported them to C++ based API of TVM for fair comparison. We wrote all our kernels by ourselves. We used simple for-loops for our kernels except for the matrix multiplication kernels of tile sizes 1x8 to 12x8, for which we used AVX2 intrinsic for vectorization. These matrix multiplication tiles were also used inside other kernels, such as convolutions or linear layers.
>
> > **Issue 3** It is better to also cite Graphi, Nimble, and IOS in Section 2.
>
> Thank you for mentioning this! We will include them in our final manuscript.

---

> > ### Comment · Reviewer_yggz · 2023-08-10
> >
> > Thanks for your detailed response, looking for your final manuscript and source code of ASPEN.

---

### Official Review · Reviewer_qSHs · 2023-06-30

**Soundness:** 3 good
**Presentation:** 3 good
**Contribution:** 2 fair
**Rating:** 6
**Confidence:** 4

**Summary:**

DNN is composed of multiple computational blocks, each using different tensor operators. However, due to the nature of the computational graph, there are internal dependencies among the blocks and operators. Consequently, the synchronization barrier results in considerable overhead for modern high-parallelism hardware. This paper introduces ASPEN, which aims to mitigate the synchronization barriers between operators and to uncover parallel computation opportunities across operators. ASPEN consists of two stages of optimizations: offline and runtime. In the offline optimization stage, an Automated Parallelism Unit (APU) is employed to convert operators into a tile-based fine-grained data-flow graph. Then, during runtime, the workloads are distributed across the hardware. The results demonstrate the efficient performance of ASPEN compared to other frameworks.

**Strengths:**

- This paper addresses a well-motivated problem and is easy to follow.
- The proposed techniques are generally making sense and effective.
- It includes a performance comparison of different model structures and significantly outperforms other frameworks for extreme deep model structures (such as GPT-2).

**Weaknesses:**

- The optimizations proposed in the paper appear heuristic and do not guarantee optimal performance.
- The evaluation is not thorough, lacking some details on memory consumption and comparisons with other state-of-the-art frameworks, such as TASO, ONNXRuntime, and oneDNN.
- It is unclear how to remove the barrier in the APU optimization. Does this involve a trade-off in memory consumption by loading multiple weights into parallel units?

**Questions:**

- Algorithm 1 appears to be ad-hoc and heuristic, as it relies on the status of node execution. However, this heuristic cannot guarantee optimality and is not always effective. Have the authors considered other algorithms?
- Have the results from TVM been tuned?
- How does the memory consumption compare to other baselines? Do the optimizations incur any additional overhead?
- While it is good for this paper to compare with different frameworks, including DNN primitive framework and compiler-based framework, TensorFlow is a general training framework, whereas ONNXRuntime and oneDNN generally perform better in the inference domain. It would be better if the authors could provide a performance comparison for these frameworks.
- Is the improvement in speed mainly due to the Convolution and GEMM kernels?

**Limitations:**

Yes

---

> ### Author Rebuttal · Authors · 2023-08-09
>
> Thank you for your valuable comments and feedback! In this response, we will address your individual concerns in addition to the general response. We hope this clarifies any concerns about our work.
>
> ---
> > **Issue 1** The optimizations/algorithm appear to be ad-hoc and heuristic.
>
> Our work aims to present a dynamic DNN scheduling and execution approach that can increase parallel resource utilization during runtime using tile-based opportunistic parallelism. Therefore, the main algorithm (Algorithm 1) is not designed to optimize the execution of certain DNNs or environments. Instead, it focuses on enabling opportunistic parallelism and dynamically increasing resource utilization by continuously scheduling any newly available computations during runtime. This strategy permits each computation resource to accommodate as many computations as the DNN offers throughout the runtime. By keeping every resource maximally utilized, the algorithm can collectively achieve high throughput, regardless of the DNN or hardware used.
>
> We agree that maximal utilization of the resources does not guarantee optimality, and there exists room for improvement in our algorithm. For instance, we find that some tiles are more important than others, and prioritizing them can accelerate the execution, due to the dependency patterns in the DNN formed from pooling and strided layers. In this work, however, in order to focus on providing a general solution that first enables the novel approach of tile-based opportunistic parallelism, we left optimizations for future work. Nonetheless, we will include more details on the optimality of our algorithm, and other dynamic scheduling approaches that can lead to optimality in executions.
>
> > **Issue 2** Removal of the barrier in the APU optimization is unclear. Does this involve loading multiple weights?
>
> The key to removing synchronization barriers is in both the tile-based decomposition of DNNs in the APU and the dynamic dependency tracking in the runtime. Partitioning each operator and creating tile-based DNN which allows each tile to be managed as a graph node. Dependency is atomically updated for each node in runtime, allowing resources to check dependency states without synchronization. Through this, ASPEN removes the need for synchronization barriers and allows for much higher parallel resource utilization.
>
> This also means that there is loading of multiple weights, as the computation remains the same. The details of ASPEN memory usage are further explained in the next response. In our final manuscript, we will make it clear that our contribution is not an optimization or trade-off on the existing algorithms, but a new approach to scheduling and parallelization of DNN computation.
>
> > **Issue 3** How does memory consumption compare? Do the optimizations incur any additional overhead?
>
> We must first make it clear that ASPEN improves the parallel scheduling of DNN computations and does not alter the computation process of the DNN. Additional memory required by ASPEN is for creating tile-based graphs, and its nodes act only as instructions to compute the given tile location in a tensor. As such, it only holds references (pointers) to its respective tile locations and the tile locations referenced by the nodes are still a part of the original input/output tensors, as in the conventional operator-based approach.
>
> The memory overhead of the tile-based graph used by ASPEN is minimal, as each node requires only ~100 bytes of memory. For example, in the ResNet-50 batch 1 case, ASPEN graph requires 836 kilobytes of additional memory, which is minuscule compared to the weights of ResNet-50, which is ~100 megabytes.
>
> Our submitted supplementary material of ASPEN may use a relatively larger amount of memory, due to the more relaxed memory allocations we used in our implementation. This is not reflective of the memory use required by the ASPEN algorithm itself. We will update our code to manage its memory as tightly as possible.
>
> In our final manuscript, we will include explanations of how data are stored and managed in ASPEN, along with a detailed decomposition of ASPEN memory usage over different devices and DNNs. We also plan to fully release the ASPEN source code, so any details of ASPEN and its inner workings would become transparent.
>
> > **Issue 4** Are results from TVM tuned? It would be better if there are comparisons against more frameworks.
>
> We use AutoTVM, following the instructions provided in the official TVM documentation. We compile our model in Python and export the model to C++ based TVM API for a fair comparison. We tried to include as many comparison baselines as possible, available in the C/C++ domain. However, some inference solutions include optimizations such as caching, approximations, or data modifications, which are largely orthogonal to the parallelism and scheduling contributions of ASPEN and make them unfair to compare to.
>
> However, we plan to include the suggested frameworks such as ONNXRuntime and oneDNN in our final manuscript, as these frameworks provide options to adjust the level of optimization and allow for a fair comparison.
>
> > **Issue 5** Is the speedup mainly from the computation kernels?
>
> Our improvement comes from the higher device utilization from the removal of synchronization barriers, increased parallelism, and data reuse from depth-first node execution, which allows better parallel scaling and increased per-resource throughput. As explained in the general response, our tile-based kernels are less performant than those of existing operator-based solutions.
>
> We will include a more detailed decomposition of the performance contributions of ASPEN, including the performance of ASPEN kernels. However, it must be noted that contributions of ASPEN such as increased parallelism over operator boundaries, asynchronous interleaving of scheduling and computation, or data reuse from depth-first execution are most prominent in full end-to-end execution.

---

> > ### Author Response · Authors · 2023-08-11
> >
> > We have made a slight typo in Issue 2, in the first sentence of the second paragraph.
> >
> > > This also means that there is loading of multiple weights, as the computation remains the same.
> >
> > should be corrected to
> >
> > > This also means that there is **no** loading of multiple weights, as the computation remains the same.
> >
> > Sorry if this has caused any confusion.

---

> > ### Comment · Reviewer_qSHs · 2023-08-19
> >
> > Thank you for your further explanation, which has partially resolved my concerns. The author also promised to provide further evaluation and clarification in the final manuscript. I will raise my rating accordingly.

---

### Official Review · Reviewer_b91G · 2023-07-03

**Soundness:** 3 good
**Presentation:** 3 good
**Contribution:** 3 good
**Rating:** 6
**Confidence:** 2

**Summary:**

In this paper, authors proposed ASPEN, a parallel computation solution for DNNs, which utilizes a new class of parallelism for DNNs, namely opportunistic parallelism, to dynamically locate and execute any parallel computation opportunities during runtime. More specifically, the authors have presented three main key points in ASPEN:
(1) a tile-based graph partitioning unit that transforms operator-based DNN dataflow graphs into tile-based dataflow graphs unlocking rich parallel computation opportunities,(2) a distributed scheduling algorithm that enables each resource to asynchronously track and compute a distinct computation path without encountering any data hazards or race conditions, and (3) a highly concurrent data structure that facilitates asynchronous information exchange among parallel resources.
The evaluation of ASPEN on various CNN and transformer-based model inferences on CPU have shown performance gains.

**Strengths:**

1. The paper is clear and organized. The authors presented ASPEN in three key components: the Automated Parallelism Unit (APU), the Distributed Scheduling Engine (DSE), and the Ready Pool. each component is well introduced and analyzed. The overall workflow of ASPEN is clear.
2. ASPEN has shown strong performance on various DNN architectures, achieving speedups up to 6.2× against TensorFlow and 4.3× against TVM (GPT-2 S1024 B1).

**Weaknesses:**

1. As mentioned in the paper, ASPEN is targeted at CPUs. The motivation is not quite strong if only applicable to CPUs.


**Questions:**

N/A

**Limitations:**

The hardware platform is limited to CPU, which is the biggest limitation for the proposed ASPEN.

---

> ### Author Rebuttal · Authors · 2023-08-09
>
> Thank you for your valuable comments and feedback! In this point-to-point response, we will address and hopefully clarify your concern with our work.
>
> ---
>
> > **Issue 1** The hardware platform is limited to CPUs.
>
> As explained in detail in the general response, ASPEN is not necessarily limited to CPUs. ASPEN provides a general concept and solution to enabling a novel DNN scheduling and execution system that allows higher parallelism resource utilization, not confined to certain hardware architecture. However, due to its novelty, we were unable to find a suitable computation backend for GPUs. Consequently, we were only able to provide evaluation results based on CPUs. Fortunately, we find that tile-based kernels are becoming more widely available, and therefore we expect the GPU backend for ASPEN to be soon available.
> We will make this clearer in our final manuscript and explain in more detail how ASPEN is applicable to other hardware architecture, as explained in the general response.
>
> ---
>
> Thank you again for your valuable feedback and suggestions on our work!

---

### Official Review · Reviewer_zpkQ · 2023-07-07

**Soundness:** 3 good
**Presentation:** 3 good
**Contribution:** 3 good
**Rating:** 6
**Confidence:** 3

**Summary:**

Inference performance is one of the key metrics driving the commercial adoption and integration of modern DNNs into user-facing applications. In the paper, the authors propose a framework, ASPEN, to improve the inference performance of DNNs by exploiting a novel strategy to extract maximal parallelism, called opportunistic parallelism, during the forward pass. Opportunistic parallelism dynamically locates and executes ready units of computational maximally based on the minimization of the number of synchronization barriers introduced by different operators. Synchronization barriers are removed based on a tile-wise decomposition of the input data and weights to produce parallel execution sequences capable of taking advantage of more hardware. Orchestration of the computational work is performed by 2 processes, the automated parallelism unit (APU), lowers the input graph from an operator-centric synchronization strategy to a tile-wise dataflow graph suitable for parallel execution with a small number of synchronization barriers. The lowered dataflow graph is then placed on the compute units by the distributed scheduling engine to assign available processors to units of work as the availability, or ready state, of the dataflow graph dictates. Based on these optimizations the authors demonstrate significant performance improvements over competing inference engine systems as the number of cores is increased.

**Strengths:**

- The significance of the work to the community is clear as exploiting the maximal efficiency of hardware during the deployment of DNNs is of paramount importance for industrial applications.
- The lowering of the operator-oriented DNN to a dataflow representation that exposes ample parallelism is interesting and the necessity of the authors to reimplement a large number of the lower primitives to achieve this feat speaks to the novelty of the approach. This also brings into question the current strategies employed by existing libraries/frameworks to execute DNNs and whether they should be extended to allow for a finer level of granularity required to facilitate the parallelism exposed by ASPEN.
- The ASPEN dataflow graph combined with the DSE runtime system is shown to achieve substantial performance over competing implementations for a number of different DNNs and multiple CPU architectures.

**Weaknesses:**

- The evaluation section focuses on comparisons with other CPU architectures and ignores providing any data on competing implementations optimized for GPUs.
- It is not clear the insights regarding the tensor decompositions and the resulting dataflow graph would easily generalize to GPU architectures or for the training phase. These concerns are acknowledged by the authors.

**Questions:**

- Though the results presented are CPU-based I wonder if comparing with GPU inference results, based on something like TensorRT, would be useful?

**Limitations:**

The limitations were adequately addressed by the authors in the text.

---

> ### Author Rebuttal · Authors · 2023-08-09
>
> Thank you for your valuable comments and feedback! In this response, we will address your individual concerns one by one in addition to the general response. We hope this clarifies your concerns about our work.
>
> ---
>
> > **Issue 1** The evaluation only focuses on CPUs and does not contain GPU results or comparisons against GPU inference results.
> It is also not clear if the tile-based dataflow graph would easily generalize to GPUs.
>
> As explained in the general response, due to the unique nature of ASPEN’s approach, a suitable computation backend for GPU is not currently available. Consequently, our evaluation results are presently confined to CPUs. However, we have observed a growing availability of tile-based kernels, and therefore we expect that the performance benefits of ASPEN on GPUs will soon be corroborated, which is rather obvious according to the principles of ASPEN.
>
> Also, the tile-based dataflow graph and its execution of ASPEN is generalizable to GPUs, as operator-based GPU kernels already use tile-based parallel executions internally. The asynchronous per-tile kernel launches of ASPEN can be efficiently achieved using CUDA Streams and CUDA Events API, by assigning different CUDA Streams to each DSE and enabling asynchronous tile scheduling and dependency updates with CUDA Events. By constantly keeping scheduled kernels in the CUDA Stream queues through opportunistic parallelism, ASPEN can fully utilize the SMs of the given GPU through the whole DNN execution. Also, ASPEN provides additional benefits such as the interleaving of host-device data movement and tile execution, and better utilization of memory bandwidth.
>
> It is important to note that GPU implementation results would not make any change to the core logic and design of ASPEN, implying that our contributions in this work are orthogonal to the results.
>
> > **Issue 2** Will tile-based dataflow graphs easily generalize to DNN training?
>
> We also expect that ASPEN can be readily applicable to DNN training, given that the backward propagation in DNNs primarily involves matrix multiplication, which ASPEN already supports. Gradient computations can be partitioned into tiles and the dependencies between each gradient tiles can be encoded into a graph, to create an ASPEN graph for DNN training. This graph can be fed into the ASPEN runtime to automatically perform forward and backward propagation without any modification in the ASPEN runtime. In Section 5 (Limitations and Discussions), we explain that existing approaches may use higher batch sizes to compensate for the limited parallelism of operator-based approaches, but this does not imply that ASPEN’s efficacy reduces in training. Rather, ASPEN enables additional parallelism in situations where increasing the batch size is unfavorable or impossible, such as training with limited device memory or datasets.
>
> In order to improve the readability and clarity, we will be updating Section 5 with more details regarding GPU execution and DNN training in our final manuscript.
>
> ---
>
> Thank you again for your valuable feedback and suggestions on our work!

---

> > ### Comment · Reviewer_zpkQ · 2023-08-19
> >
> > I would like to thank the authors for their thorough responses to the weaknesses I outlined in my original review. Based on their response to my review and the comments made by other reviewers I am increasing my score accordingly.

---

### Author Rebuttal · Authors · 2023-08-09

General Response
---

Thank you for taking your time to review our paper! In this response, we will explain the essential value of ASPEN and the reasons behind our selection of evaluations. Then, we will clarify individual questions and concerns one by one.

As an exploratory work on fine-grained dynamic parallelism of DNNs, ASPEN aims to provide insights and solutions to the novel computation approach of applying tile-based opportunistic parallelism to DNNs. As explained in the paper, available DNN computations are scheduled in two hierarchical layers of inter-operator and intra-operator computations using a dataflow graph of operators. However, this two-layer approach limits the available parallelism within each operator in the form of synchronization barriers. This limitation in parallelism has been identified as a major bottleneck in resource utilization by previous studies such as Rammer (Ma et al., OSDI 2020).

As described in Section 2, many solutions have been proposed to alleviate this issue. However, they still rely on the use of operators, which inevitably limits parallelism to some extent. On the other hand, ASPEN opts to take a completely new approach by removing the use of operators entirely and dynamically scheduling any parallel tiles that are available for computation. This effectively combines the separate inter- and intra-operator computation spaces into a single unified space of tile-based dataflow graph. This approach has yet been explored in the literature and enables the complete removal of the limited parallelism caused by the operator-based approach. In this framework, any computation tile can be assigned to a computation resource as soon as it becomes available, independent of which operator it belongs to, leveraging the concept of opportunistic parallelism.

Unfortunately, due to the uniqueness of our work, only a limited array of existing software infrastructure is available, curtailing its potential for the incorporation of tile-based computation kernels. For this reason, we wrote our tile-based kernels in C and left the optimizations to GCC in our evaluations on CPUs. While these kernel implementations may not be as finely optimized as the existing high-performance computation libraries, they still allow us to demonstrate the capability of ASPEN’s novel parallelism and scheduling approach in achieving substantial parallel resource utilization on CPUs. However, it is worth noting that such kernel generation is not applicable to less programmable hardware such as GPUs, as their code is much more hardware dependent with proprietary software stacks, and their kernel compilers are not versatile enough to be applicable to tile-based kernels. As a result, our demonstration of the efficacy of ASPEN is limited to its maximal available scope (as of now).

Nonetheless, we believe that with the appropriate computation kernels, the novel approach proposed by ASPEN will greatly benefit DNN executions on GPUs, for the following reasons. One of the limiting factors to a high GPU utilization is the host-device scheduling and communication overhead in both data movement and kernel launches, as reported in studies such as Rammer and Nimble (Kwon et al., NeurIPS 2020). The dynamic and asynchronous nature of ASPEN allows for the seamless interleaving of data movement, kernel launches, and tile executions, enabling concurrent scheduling and processing. As a result, ASPEN greatly reduces the overhead of host-device communications. This asynchronous nature also means that the computing resources are at different stages of kernel execution during computation, which leads to the distribution of memory access requests across the temporal domain. This decreases the pressure on the shared memory bus and allows better utilization of the memory system, which mitigates the limitations in memory bandwidth that DNN executions on GPUs often face, as mentioned in studies such as Alpa (Zheng et al., OSDI 2022) and Welder (Shi et al., OSDI 2023).

Furthermore, we find that tile-based GPU kernels for DNNs are becoming available soon. To be specific, we find that Welder, a memory optimization work released a few weeks ago (July 2023), constructs a tile-based dataflow graph during its offline optimization stage, which can be used as an ASPEN graph with minor modifications. Although Welder still uses conventional operator-based execution during runtime, thus keeping ASPEN’s contributions untouched, we find that its GPU kernels can be decomposed to its offline tile-based form and be modified into a computation backend suitable for ASPEN. Leveraging these kernels, an effective extension of ASPEN for (Nvidia) GPUs is feasible using CUDA Streams and CUDA Events API. A separate CUDA Stream allocated to each DSE allows kernels launched by different DSEs to run concurrently, and DSEs can asynchronously check the completion of a kernel and update its child nodes using CUDA Events. Regardless, GPU support for ASPEN would not alter the core logic and design of ASPEN in any way, and therefore the contributions and benefits of ASPEN presented in our paper remain intact.

Tile-based understanding of DNNs has gained significant traction in recent years, as it provides a holistic view of the DNN and allows for a finer-grained analysis and management of both computation and data, which enables contributions that were previously impossible with the operator-based dataflow graphs. ASPEN is the first to explore the benefits of tile-based DNNs during runtime. Unlike previous tile-based works which focused on offline analysis and optimizations such as graph compilers (Roller, Zhu et al., OSDI 2022) or memory usage (Welder), ASPEN focuses on the benefits of tile-based DNNs on parallel resource utilization during execution using opportunistic parallelism. We hope that this clarifies the core value of ASPEN as an exploratory work providing insights and algorithms for tile-based dynamic scheduling and computation of DNNs.

---

### Decision · Program_Chairs · 2023-09-21

**Decision:**

Accept (poster)

**Comment:**

This paper presents a framework for exploiting fine-grained parallelism to improve the performance of neural networks, particularly on CPUs. All of the reviewers supported acceptance and agreed that the results were promising. There seems to still be a long way to go in fully proving out the approach in this paper in practice (e.g., for multiple CPUs, larger batch sizes, single and multiple GPUs, etc.), but the promise of the work merits publication in my mind. I expect the authors (and many others) to build on this work toward making it practically impactful.